

# Dualities and loops on squashed $S^3$

**Charles Thull⋆**

Department of Physics and Astronomy, Uppsala University,
Box 516, SE-751 20 Uppsala, Sweden

⋆ charles.thull@physics.uu.se

## Abstract

We consider $\mathcal{N} = 4$ supersymmetric gauge theories on the squashed three-sphere with six preserved supercharges. We first discuss how Wilson and vortex loops preserve up to four of the supercharges and we find squashing independence for the expectation values of these $\frac{2}{3}$-BPS loops. We then show how the additional supersymmetries facilitate the analytic matching of partition functions and loop operator expectation values to those in the mirror dual theory, allowing one to lift all the results that were previously established on the round sphere to the squashed sphere. Additionally, on the squashed sphere with four preserved supercharges, we numerically evaluate the partition functions of ABJM and its dual super-Yang-Mills at low ranks of the gauge group. We find matching values of their partition functions, prompting us to conjecture the general equality on the squashed sphere. From the numerics we also observe the squashing dependence of the Lee-Yang zeros and of the non-perturbative corrections to the all order large $N$ expression for the ABJM partition function.

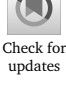

# 1   Introduction

Supersymmetric localization has been used in a variety of contexts for the exact evaluation of partition functions and loop operator expectation values in supersymmetric quantum field theories with at least two preserved supercharges [1]. In three dimensions it was first used on the round sphere [2,3]. The extension to squashed three-spheres depends on the supercharges one chooses to preserve [4,5] but can only depend on a single scalar parameter of the geometry [6]. Still, this one parameter deformation complicates evaluations and transformations of the resulting matrix models by changing hyperbolic trigonometric functions to double sine functions.

In [7], backgrounds on the squashed three-sphere which preserve six supercharges were considered.[1] The additional two supercharges were understood as an enhancement of the supersymmetry upon fine-tuning of a specific mass parameter in the previously known backgrounds with four supercharges. The advantage of this new setup is that for every $\mathcal{N} \geq 4$ gauge theory[2] on this background the partition function is independent of the squashing parameter. For ABJ(M) this squashing independence was first noticed in [9] and furnished useful relations on the derivatives of the free energy. This squashing independence of the ABJM partition function has also been discussed from a 4d index [10].

In this paper we consider loop operators in $\mathcal{N} = 4$ theories on the squashed sphere preserving six supercharges. To see the effects of the additional two supercharges, we start with the squashed-sphere with four supercharges and its $\frac{1}{2}$-BPS loop operators. Once we enhance the supersymmetry to six supercharges these operators generally do not preserve the additional supercharges and are therefore $\frac{1}{3}$-BPS. However exactly two of the great circles support $\frac{2}{3}$-BPS loops. A Wilson loop living on one of these two circles preserves the same four supercharges as the vortex loop living on the other one. We will argue that $\frac{2}{3}$-BPS versions of all loop operators in [11] exist on this background. We expect that the loop operators in non-linear quivers theories as discussed in [12,13] also admit $\frac{2}{3}$-BPS versions on the squashed sphere. We leave for future work the lift to the squashed sphere of purely $\mathcal{N} = 4$ loop operators, such as those in [14] for example.

We use localization to compute the expectation values of the $\frac{1}{3}$-BPS loop operators [2,11, 15,16]. We find that once a Wilson or vortex loop operator preserves four supercharges the dependence on the squashing parameter drops out of the matrix integral in the same way as it dropped out of the partition function for six preserved supercharges.

In the vast landscape of $\mathcal{N} = 4$ theories these results on squashing independence of partition functions and loop operator expectation values apply at least to all Lagrangian theories. Examples are the $\mathcal{N} = 8$ SYM and ABJ(M) theories, as well as the theories from Hanany-

---

[1]Similar backgrounds were considered in [8].

[2]Here and in the following we will denote gauge theories by the amount of supersymmetry they have in flat-space. By contrast, for rigid supergravity backgrounds we will indicate the number of supercharges they preserve.

Witten style brane constructions [17]. In the IR many of these theories are not independent but they are interrelated by dualities, including mirror symmetry [17–19] and Seiberg-like dualities [20, 21]. A common test for such dualities is the matching of the partition functions as well as loop operator expectation values on both sides. This has been done extensively on the round sphere, *e.g.* in [22–36].

On the squashed sphere, checking dualities by equating partition functions requires more work. Seiberg-like dualities have been checked on the squashed sphere by dimensional reduction of the 4d index [37]. Relying on [38], one can even check Seiberg-like dualities in theories with chiral matter [39, 40]. However, mirror symmetry on the squashed sphere has had less success. Only linear quivers have recently been treated using a local recursive dualization algorithm based on Seiberg-like dualities [41, 42]. In this work we observe that the difficulties arise from breaking the $\mathcal{N} = 4$ supersymmetry with backgrounds that preserve only four supercharges. However, as soon as the background preserves six supercharges all matchings of partition functions and loop operator expectation values can be lifted from the round to the squashed sphere, which for mirror symmetry means we can observe the exchange of the Coulomb and the Higgs branches as well as of Wilson and vortex loops.

We show numerically that the expected matching of the partition functions between ABJM and $\mathcal{N} = 8$ SYM holds at low ranks on the squashed sphere with four preserved supercharges. We show this for a wide range of squashing parameters, Fayet-Iliopoulos coefficients and masses. Based on this we conjecture the general equality. We expect that the techniques of [42, 43] may be used to prove this relation analytically.

From the numerical data we observe surfaces in the three dimensional parameter space where the partition function goes to zero. These correspond to the Lee-Yang zeros already found on the round sphere [44, 45]. We observe from the numerical data that squashing moves them to larger FI and mass parameter values. We also compare our numerical data to the all-order large $N$ partition function of ABJM that was recently conjectured [46]. This allows us to observe the growth of the instanton corrections to the ABJM free energy under squashing. It would be interesting to understand both the Lee-Yang zeros and the non-perturbative corrections further.

This paper is structured as follows. In section 2 we review supergravity backgrounds on the squashed three-sphere with four and six preserved supercharges and localized $\mathcal{N} = 4$ partition functions. In section 3 we discuss BPS loop operators, their expectation values and squashing independence for the $\frac{2}{3}$-BPS loops. In section 4 we discuss how our results allow to lift tests of dualities from the round to the squashed sphere as well as the numerical results for matching the partition functions of ABJM and $\mathcal{N} = 8$ SYM.

## 2  Supersymmetry on the squashed three sphere

In this section we give a very brief review of supersymmetry on the squashed three-sphere. The squashed three-sphere has the metric [5][3]

$$ds^2_{\text{squashed}} = \frac{r_3^2}{4}\left(\frac{b + b^{-1}}{2}\right)^2 (d\psi + \cos\theta \, d\phi)^2 + \frac{r_3^2}{4}\left(d\theta^2 + \sin^2\theta \, d\phi^2\right), \tag{1}$$

---

[3]This is the same metric as the familiar squashing in [4] but we choose to preserve a different set of supersymmetries. Note that often the name squashed sphere gets used to denote the ellipsoid as well despite it having a different metric.

where $b$ is the squashing parameter and $b = 1$ is the round sphere. We use the coordinates $\theta \in [0, \pi]$, $\phi \in [0, 2\pi]$ and $\psi \in [0, 4\pi]$. We pick the frame

$$
\begin{aligned}
e^1 &= -\frac{r_3}{2} \left( \sin \psi \, \mathrm{d}\theta - \sin\theta \cos\psi \, \mathrm{d}\phi \right), \\
e^2 &= \frac{r_3}{2} \left( \cos \psi \, \mathrm{d}\theta + \sin\theta \sin\psi \, \mathrm{d}\phi \right), \\
e^3 &= -\frac{r_3}{4} \left( b + \frac{1}{b} \right) (\mathrm{d}\psi + \cos\theta \, \mathrm{d}\phi).
\end{aligned}
\tag{2}
$$

The $\mathcal{N} = 2$ Killing spinor equations are [47]

$$
\begin{aligned}
\nabla_\mu \zeta - \mathrm{i} \left( A_\mu + V_\mu \right) \zeta - H \gamma_\mu \zeta + \frac{1}{2} \varepsilon_{\mu\nu\rho} V^\nu \gamma^\rho \zeta &= 0, \\
\nabla_\mu \widetilde{\zeta} + \mathrm{i} \left( A_\mu + V_\mu \right) \widetilde{\zeta} - H \gamma_\mu \widetilde{\zeta} - \frac{1}{2} \varepsilon_{\mu\nu\rho} V^\nu \gamma^\rho \widetilde{\zeta} &= 0.
\end{aligned}
\tag{3}
$$

If the background fields take the values

$$
H = \frac{\mathrm{i}}{4 r_3} \left( b + b^{-1} \right), \qquad V = -A = \frac{1}{4} \left( b + b^{-1} \right) \left( b - b^{-1} \right) (\mathrm{d}\psi + \cos\theta \, \mathrm{d}\phi),
\tag{4}
$$

then for all constant spinors $\zeta_0, \widetilde{\zeta}_0$ we have the preserved Killing spinors [5]

$$
\zeta = e^{\frac{\mathrm{i}}{2}\Theta\sigma_3} \cdot g^{-1} \cdot \zeta_0, \qquad \widetilde{\zeta} = e^{-\frac{\mathrm{i}}{2}\Theta\sigma_3} \cdot g^{-1} \cdot \widetilde{\zeta}_0,
\tag{5}
$$

where we defined

$$
e^{\mathrm{i}\Theta} = -b, \qquad g = \begin{pmatrix} \cos\frac{\theta}{2} e^{\frac{\mathrm{i}}{2}(\phi+\psi)} & \sin\frac{\theta}{2} e^{\frac{\mathrm{i}}{2}(\phi-\psi)} \\ -\sin\frac{\theta}{2} e^{-\frac{\mathrm{i}}{2}(\phi-\psi)} & \cos\frac{\theta}{2} e^{-\frac{\mathrm{i}}{2}(\phi+\psi)} \end{pmatrix}.
\tag{6}
$$

Thus we have a total of 4 preserved supercharges and the preserved superalgebra is $\mathfrak{su}(2|1) \oplus \mathfrak{u}(1)$.

If we take this as a background for an $\mathcal{N} = 4$ theory, only the diagonal $U(1)_{diag}$ subgroup of the full R-symmetry group $SU(2)_C \times SU(2)_H$ is preserved as an R-symmetry while the axial subgroup $U(1)_{ax}$ takes the role of a flavor symmetry. This $U(1)_{ax}$ can be coupled to a background vector multiplet $(A, \lambda = 0, m_*, D)$ where $m_*$ is a mass parameter and we will refer to it as the *special mass*. Such a background vector multiplet has to take the form

$$
D = -m_* \frac{H}{r_3}, \qquad A = \frac{2\mathrm{i} m_*}{b + b^{-1}} V,
\tag{7}
$$

to preserve the above 4 supercharges on $S_b^3$.

For $m_* = \pm \mathrm{i} \frac{b - b^{-1}}{2}$ the $U(1)_{ax}$ background vector multiplet combines with the $\mathcal{N} = 2$ background supergravity multiplet to give an $\mathcal{N} = 4$ background supergravity multiplet with two additional preserved supercharges and an enhanced superalgebra $\mathfrak{su}(2|1) \oplus \mathfrak{su}(1|1)$. The two additional Killing spinors are

$$
m_* = \mathrm{i} \frac{b - b^{-1}}{2}: \qquad \zeta' = \begin{pmatrix} 1 \\ 0 \end{pmatrix}, \qquad \widetilde{\zeta}' = \begin{pmatrix} 0 \\ 1 \end{pmatrix},
\tag{8}
$$

$$
m_* = -\mathrm{i} \frac{b - b^{-1}}{2}: \qquad \zeta' = \begin{pmatrix} 0 \\ 1 \end{pmatrix}, \qquad \widetilde{\zeta}' = \begin{pmatrix} 1 \\ 0 \end{pmatrix}.
\tag{9}
$$

We will refer to these values of $m_*$ as the *enhancement points* for the supersymmetry.[4]

---

[4]See the appendix of [7] for the 3d conformal supergravity version of this $\mathcal{N} = 4$ background. [8] discusses similar squashed three-sphere backgrounds with more than four preserved supercharges.

In [7] we discussed the twisted dimensional reduction of the 4d index on $S^1 \times S^3$ in the context of additional preserved supercharges.[5] Let us highlight one aspect of this discussion which will simplify our work later. On $S^1 \times S^3$ the twist corresponds to a non-trivial rotation on three-sphere as we go around the circle. This inserts a factor

$$\exp\left(-4\pi \frac{b-b^{-1}}{b+b^{-1}} \frac{r_1}{r_3} j_2\right), \tag{10}$$

in the index where $j_2$ is the charge for the $SU(2) \subset SO(4)$ subgroup of the rotations on $S^3$ that leave four Killing spinors $\zeta, \tilde{\zeta}$ invariant. This rotation also acts on the circles that the $\frac{2}{3}$-BPS loop operators lie on. Hence when computing an index for a 1d theory on these circles we will have to remember to insert this factor.

Once we couple an $\mathcal{N} = 4$ gauge theory to the squashed sphere background with at least four supercharges, its partition funciton can be computed using supersymmetric localization. The result is an integral over the Cartan subalgebra of the gauge group with measure $d\hat{\sigma}^{\text{rank } G} \prod_{\alpha \in \Delta_+} \langle \hat{\sigma}, \alpha \rangle^2$ and additional insertions dependent on the content of the theory. For each vector- or hypermultiplet we have to include the one-loop determinants:

$$Z_{\mathcal{N}=4}^{\text{vec}}(m_*) = S_2\left(\frac{Q}{2} - im_*\right)^{\text{rank } G} \prod_{\alpha \in \Delta_+} \frac{S_2(\pm i\langle \hat{\sigma}, \alpha \rangle; b, b^{-1}) S_2(\frac{Q}{2} - im_* \pm i\langle \hat{\sigma}, \alpha \rangle; b, b^{-1})}{\langle \hat{\sigma}, \alpha \rangle^2}, \tag{11}$$

$$Z_{\mathcal{N}=4}^{\text{hyp}}(m_*, m) = \prod_{\rho \in \mathcal{R}} S_2\left(\frac{Q}{4} - \frac{im_*}{2} \pm i(\langle \hat{\sigma}, \rho \rangle + m); b, b^{-1}\right)^{-1}, \tag{12}$$

where $S_2(x; b, b^{-1})$ is the double sine function defined in Appendix A. We use the shorthand notation $S_2(x \pm y) = S_2(x+y)S_2(x-y)$. Once we go to one of the supersymmetry enhancement points, the one-loop determinants simplify significantly

$$Z_{\mathcal{N}=4}^{\text{vec}}(m_*) = \begin{cases} b^{-\text{rank } G} \prod_{\alpha \in \Delta_+} \frac{\sinh(\pi b^{-1}\langle \hat{\sigma}, \alpha \rangle)^2}{\langle \hat{\sigma}, \alpha \rangle^2}, & m_* = i\frac{b-b^{-1}}{2}, \\ b^{\text{rank } G} \prod_{\alpha \in \Delta_+} \frac{\sinh(\pi b\langle \hat{\sigma}, \alpha \rangle)^2}{\langle \hat{\sigma}, \alpha \rangle^2}, & m_* = -i\frac{b-b^{-1}}{2}, \end{cases} \tag{13}$$

$$Z_{\mathcal{N}=4}^{\text{hyp}}(m_*, m) = \begin{cases} \prod_{\rho \in \mathcal{R}} \frac{1}{\cosh(\pi b^{-1}(\langle \hat{\sigma}, \rho \rangle + m))}, & m_* = i\frac{b-b^{-1}}{2}, \\ \prod_{\rho \in \mathcal{R}} \frac{1}{\cosh(\pi b(\langle \hat{\sigma}, \rho \rangle + m))}, & m_* = -i\frac{b-b^{-1}}{2}. \end{cases} \tag{14}$$

Important for later is the b-dependence of these expressions at the enhancement points. A Chern-Simons term at level $k$ inserts the factor

$$\exp\left(-i\frac{\pi}{2} k \operatorname{tr}(\hat{\sigma}^2)\right), \tag{15}$$

into the matrix model, while a Fayet-Iliopoulos (FI) term with parameter $\xi$ inserts

$$\exp(2\pi i \xi \operatorname{tr}(\hat{\sigma})). \tag{16}$$

## 3 Wilson and vortex loops

Loop operators are important observables in gauge theories. In this section we discuss loop operators on the squashed three-sphere that preserve two or four of the six supercharges of the $\mathcal{N} = 4$ squashed sphere discussed in the previous section. For the operators preserving at least two supercharges we find their expectation values in terms of a matrix model integral and for the $\frac{2}{3}$-BPS loops we show that the squashing dependence drops out.

---

[5]See [10] for an extension to the context of 4d $\mathcal{N} = 3$ theories and ABJM theory.

## 3.1 Wilson loops

A Wilson loop for a gauge symmetry in a three-dimensional $\mathcal{N} = 2$ theory corresponds to an insertion into the path integral of the path ordered exponential

$$\text{tr}_{\mathcal{R}} \mathcal{P} \exp\left( i \oint_{\gamma} (A - i x \sigma |d\gamma|) \right), \tag{17}$$

where $A$ is the gauge field, $\sigma$ is the scalar in the $\mathcal{N} = 2$ vector multiplet, $\gamma$ is a closed path in our space, $\mathcal{R}$ a representation of the gauge group and $x$ some constant. For BPS-loops the path $\gamma$ is constrained and $x$ is fixed. To preserve a pair of supercharges $\xi, \widetilde{\xi}$, the SUSY variation of the integrand should vanish, hence the following equation has to hold

$$\begin{aligned} 0 &= \delta_{\xi}(A(K) - i x \sigma |K|) \\ &= -i(\xi \gamma_{\mu} K^{\mu} - x|K|\xi)\widetilde{\lambda} - i(\widetilde{\xi}\gamma_{\mu}K^{\mu} + x|K|\widetilde{\xi})\lambda, \end{aligned} \tag{18}$$

where $K = \frac{d\gamma}{ds}$ is the tangent to the path $\gamma$ and we have used the SUSY conventions from [47]. The terms in both sets of parentheses have to vanish independently and it is obvious that this requires $K$ to be related to the Killing spinors. The natural choice for $K$ is the Killing vector

$$K_{\mu} = \xi \gamma_{\mu} \widetilde{\xi}, \tag{19}$$

such that by construction

$$\xi(\gamma_{\mu}K^{\mu} - (\xi\widetilde{\xi})) = 0, \qquad \widetilde{\xi}(\gamma_{\mu}K^{\mu} + (\xi\widetilde{\xi})) = 0.$$

Thus in order for the line operator to preserve the two supercharges we must set $x = \frac{(\xi\widetilde{\xi})}{|K|} \in \{-1, 1\}$. Concretely, on the squashed three-sphere we can choose the spinors $\zeta_0 = (1, 0)$ and $\widetilde{\zeta}_0 = (0, -1)$ to define $\zeta, \widetilde{\zeta}$ in (5).[6] Then the Killing vector takes the form

$$K = \frac{2}{r_3}\left(b - b^{-1}\right)\partial_{\psi} + \frac{2}{r_3}\left(b + b^{-1}\right)\partial_{\phi}, \tag{20}$$

and we see that this gives a supersymmetric Wilson line with $x = 1$. At fixed but generic $\theta$ the loop is closed only for $b^2$ rational. The exceptions are at $\theta = 0$ and $\pi$ where the loop closes for all $b$. Any additional Killing spinor $\zeta'$ or $\widetilde{\zeta}'$ from (5) that is preserved in the presence of this line has to satisfy

$$\zeta'(\gamma_{\mu}K^{\mu} - |K|) = 0, \qquad \widetilde{\zeta}'(\gamma_{\mu}K^{\mu} + |K|) = 0. \tag{21}$$

However the only solutions of the form (5) are proportional to $\zeta, \widetilde{\zeta}$, confirming that the Wilson lines are necessarily $\frac{1}{2}$-BPS on the squashed sphere with four supercharges. This can also be seen from the superalgebra $\mathfrak{su}(2|1)$, as a line operator breaks the rotational $\mathfrak{su}(2)$ subalgebra down to $\mathfrak{u}(1)$.

To find loops with additional preserved Killing spinors, we need the enhanced supersymmetry with its superalgebra $\mathfrak{su}(2|1) \oplus \mathfrak{su}(1|1)$. From the previous paragraph we know that a line operator breaks $\mathfrak{su}(2|1)$ to $\mathfrak{su}(1|1)$. For a line to preserve this together with a second $\mathfrak{su}(1|1)$ the Killing vectors for both superalgebras have to be parallel on the line. Explicitly, assume we have tuned the special mass $m_*$ to $i\frac{b-b^{-1}}{2}$ and thus have the additional conserved supercharges $\zeta' = (1, 0), \widetilde{\zeta}' = (0, 1)$ on the squashed sphere. The Killing vector for the new supercharges is

$$\zeta'\gamma_{\mu}\widetilde{\zeta}'dx^{\mu} = -e^3. \tag{22}$$

---

[6]We focus on this pair because it is the one used for the localization computation in [5]

This is parallel to $K_\mu = \zeta \gamma_\mu \widetilde{\zeta}$ in (20) only on the circles at $\theta = 0, \pi$. Assuming $A, \sigma$ are part of a standard $\mathcal{N} = 4$ vector multiplet, the equations (21) have to hold as well to preserve $\zeta', \widetilde{\zeta}'$. This restricts the loop preserving four supercharges to lie at $\theta = \pi$. For a twisted vector multiplet the signs in equations (21) are flipped and thus the $\frac{2}{3}$-BPS Wilson loop has to lie a $\theta = 0$. If we had chosen the other enhancement point $m_* = -\mathrm{i}\frac{b-b^{-1}}{2}$, the allowed locations for the standard and twisted $\mathcal{N} = 4$ vector $\frac{2}{3}$-BPS Wilson loops would be reversed.

### 3.1.1 Evaluating Wilson loop expectation values

Let us now compute the expectation values of the BPS Wilson loops on the squashed three-sphere. All loops preserve at least two supercharges, so we can use supersymmetric localization and evaluate the Wilson loop on the localization locus, where $A = 0$ and $\sigma = \frac{2}{Qr_3}\hat{\sigma}$ for $Q = b + \frac{1}{b}$, with $\hat{\sigma}$ the Coulomb branch parameter. Thus, we insert

$$\mathrm{tr}_{\mathcal{R}} \exp\left(\frac{2}{Qr_3}\hat{\sigma}\ell(\theta)\right),\qquad(23)$$

into the matrix model integrand, where $\ell(\theta)$ is the length of the closed curve. For a generic closed loop, we have $b^2 = \frac{m}{n}$, with $m, n$ relatively prime integers. Then the length of the loop is

$$\ell(\theta) = 2\pi \frac{r_3}{2} bn|K|,\qquad |K| = Q\left(b\cos^2\frac{\theta}{2} + b^{-1}\sin^2\frac{\theta}{2}\right).\qquad(24)$$

The loops at $\theta = 0$ and $\pi$ always close and are of special interest to us as they can be $\frac{2}{3}$-BPS. For the loop at $\theta = 0$ we find

$$\ell(0) = 2\pi \frac{r_3}{2}|K(0)| = 2\pi \frac{Qr_3}{2}b.$$

Therefore this Wilson loop inserts

$$\mathrm{tr}_{\mathcal{R}} \exp(2\pi b\hat{\sigma}),\qquad(25)$$

into the matrix model. We know that this loop preserves the additional supercharges at $m_* = -\mathrm{i}\frac{b-b^{-1}}{2}$. If we now look at the 1-loop contributions (13) and (14) to the partition function at this enhancement point, we see that all contributions to the matrix model depend on the combination $b\hat{\sigma}$. Thus the $b$-dependence drops out by rescaling the Coulomb branch parameter. The same story holds for $\theta = \pi$ with $b$ replaced by $b^{-1}$, namely:

$$\langle W_{\mathcal{R}}(\theta = 0)\rangle\left(b; m_* = -\mathrm{i}\frac{b-b^{-1}}{2}\right) = \langle W_{\mathcal{R}}(\theta = 0)\rangle(1; m_* = 0),\qquad(26)$$

$$\langle W_{\mathcal{R}}(\theta = \pi)\rangle\left(b; m_* = \mathrm{i}\frac{b-b^{-1}}{2}\right) = \langle W_{\mathcal{R}}(\theta = \pi)\rangle(1; m_* = 0).\qquad(27)$$

These results on the squashed three sphere match to the ellipsoid [48].

## 3.2 Abelian vortex loops

A vortex loop operator for a global abelian symmetry can be constructed as follows [15]: First, couple the corresponding current $j$ to a dynamical field $A_1$, then introduce a CS term for $A_1$ at level $k$. The resulting theory has an extra topological $U(1)_T$ symmetry for which the current is $\star dA_1$. Then couple this current to a another dynamical field $A_2$ and introduce a Wilson loop for $A_2$. The resulting path integral has the form

$$\int \mathcal{D}[\text{fields}] \int \mathcal{D}A_2 \int \mathcal{D}A_1 \ \exp\left(\mathrm{i}\alpha \int_\gamma A_2\right) e^{\mathrm{i}\int\left(\frac{1}{\pi}A_2\wedge dA_1 - \frac{k}{4\pi}A_1\wedge dA_1 - \frac{1}{\pi}A_1\wedge\star j\right)}e^{-S(\text{fields})}.\qquad(28)$$

Performing the $A_2$ integral introduces a $\delta$-function in field space forcing $\mathrm{d}A_1 = \alpha \delta_\gamma^{(2)} = \mathrm{d}A_\alpha$. Thus we end up with the following insertion in the path integral:

$$\exp\left(\frac{\mathrm{i}}{\pi} \int A_\alpha \wedge \star j - \mathrm{i}\frac{k}{4\pi} \int A_\alpha \wedge \mathrm{d}A_\alpha\right). \tag{29}$$

$A_\alpha$ introduces a non-trivial monodromy around the path $\gamma$ for all operators charged under the corresponding global symmetry. The quadratic insertion captures the self linking number of the loop.

For supersymmetric theories, in order to preserve some supercharges, we should replace currents with current multiplets and gauge fields with vector multiplets. The $\mathcal{N} = 2$ super-CS term is then [47]

$$\frac{k}{4\pi} \int \left(\mathrm{i}A_1 \wedge \mathrm{d}A_1 + 2(\mathrm{i}\widetilde{\lambda}_1 \lambda_1 - D_1 \sigma_1)\sqrt{g}\mathrm{d}^3 x\right), \tag{30}$$

and the $\mathcal{N} = 2$ preserving coupling of the vector multiplet $(A_2, \lambda_2, \widetilde{\lambda}_2, \sigma_2, D_2)$ to the extra topological $U(1)_T$ symmetry is

$$\frac{\mathrm{i}}{\pi} \int A_1 \wedge \mathrm{d}A_2 - \frac{1}{\pi} \int \mathrm{d}^3 x \sqrt{g} \left(\sigma_1 D_2 + \sigma_2 D_1 - \mathrm{i}\lambda_1 \widetilde{\lambda}_2 - \mathrm{i}\widetilde{\lambda}_1 \lambda_2\right). \tag{31}$$

As discussed in the previous subsection, the Wilson loop breaks some supersymmetry and thus we can rely on our discussion from the previous section to get $\frac{1}{3}$-BPS vortex loops from $\frac{1}{3}$-BPS Wilson loops.

To get a vortex loop preserving four supercharges we have to do all the above steps with $\mathcal{N} = 4$ multiplets. The super-Chern-Simons term does not allow an $\mathcal{N} = 4$ generalization and is therefore absent for loops preserving more than two supercharges. Promoting (31) to $\mathcal{N} = 4$ multiplets requires the multiplet of $A_2$ to be a twisted vector multiplet. Thus the assignment of vortex loops to the preserved four supercharges is the same as for the twisted vector $\frac{2}{3}$-BPS Wilson loops, the loop at $\theta = 0$ preserves the additional two supercharges for $m_* = \mathrm{i}\frac{b - b^{-1}}{2}$, while the one at $\theta = \pi$ preserves those at the enhancement point $m_* = -\mathrm{i}\frac{b - b^{-1}}{2}$.

### 3.2.1 Evaluating abelian vortex loops

We can evaluate the abelian vortex loop expectation value from the localized partition function. On the localization locus the scalar in the vector multiplet satisfies $\sigma = \frac{2}{Q r_3}\hat{\sigma}$ and the auxiliary field is $D = -\sigma H$ with $H$ the background field in (4), while the gauge field and fermions vanish.

Using that the volume of the squashed three-sphere is $Q\pi^2 r_3^3$ and that $H = \frac{\mathrm{i}Q}{4 r_3}$ we can evaluate the coupling of the two vector multipets (31) and the super-Chern-Simons term (30) on the localization locus

$$\frac{k}{4\pi} \int \mathrm{d}^3 x \sqrt{g}(-2)D_1 \sigma_1 = \frac{k}{4\pi}Q\pi^2 r_3^3(-2)\frac{2}{Q r_3}\hat{\sigma}_1(-1)\frac{\mathrm{i}Q}{4 r_3}\frac{2}{Q r_3}\hat{\sigma}_1 = \frac{\mathrm{i}k}{2}\pi\hat{\sigma}_1^2, \tag{32}$$

$$-\frac{1}{\pi} \int \mathrm{d}^3 x \sqrt{g}(\sigma_1 D_2 + \sigma_2 D_1) = -\frac{1}{\pi}Q\pi^2 r_3^3 2\frac{2}{Q r_3}\hat{\sigma}_1(-1)\frac{\mathrm{i}Q}{4 r_3}\frac{2}{Q r_3}\hat{\sigma}_2 = 2\pi\mathrm{i}\hat{\sigma}_1\hat{\sigma}_2. \tag{33}$$

We add a mass term $m$ in the final expression by coupling the topological $U(1)$ symmetry to a background vector multiplet. The expectation value of the abelian vortex loop is then given

by the following integral:

$$\langle V(\theta)\rangle(b;k;\alpha;m) = \int d\hat{\sigma}_1 d\hat{\sigma}_2 e^{-2\pi i m \hat{\sigma}_2 + \frac{2}{Qr_3}\alpha \hat{\sigma}_2 \ell(\theta) + 2\pi i \hat{\sigma}_1 \hat{\sigma}_2 - \frac{ik}{2}\pi \hat{\sigma}_1^2} Z(\hat{\sigma}_1)$$

$$= \exp\left(-\frac{ik}{2}\pi\left(m + \frac{i\alpha}{Qr_3\pi}\ell(\theta)\right)^2\right) Z\left(m + \frac{i\alpha}{Qr_3\pi}\ell(\theta)\right). \qquad (34)$$

Thus a vortex loop for a global $U(1)$ simply shifts the corresponding mass parameter in the partition function. This is also found on the ellipsoid [15, 16].

We conclude this subsection by observing that the vortex loop expectation value becomes squashing independent if the loop preserves four supercharges. Namely these supercharges require $k = 0$ and the loop fixed one of the poles giving the shift of the mass parameter the correct scaling for squashing independence,

$$\langle V(\theta=0)\rangle\left(b;k=0;\alpha;bm;m_* = i\frac{b-b^{-1}}{2}\right) = Z\left(b;bm+i\alpha b;m_* = i\frac{b-b^{-1}}{2}\right)$$

$$= Z(1;m+i\alpha;m_*=0)$$

$$= \langle V(\theta=0)\rangle(1;k=0;\alpha;m;m_*=0), \qquad (35)$$

$$\langle V(\theta=\pi)\rangle\left(b;k=0;\alpha;b^{-1}m;m_* = -i\frac{b-b^{-1}}{2}\right) = \langle V(\theta=\pi)\rangle(1;k=0;\alpha;m;m_*=0). \qquad (36)$$

## 3.3 Non-abelian vortex loops

To get vortex loops for non-abelian global symmetries we take the approach of [11] and couple the three-dimensional theory to a one-dimensional theory that lives on the loop. Here we argue that this coupling can also be done on the squashed sphere while preserving the four supercharges. We refer to the original paper for details on which 1d theories should be coupled to get the desired vortex (and Wilson) loops.

On the round sphere, coupling a 1d theory on a great circle to the 3d gauge theory breaks the supersymmetry to one of two distinct supersymmetry algebras of the form $\mathfrak{su}(1|1)\oplus\mathfrak{su}(1|1)$. On the loop these algebras can be identified with the $SQM_V$ or $SQM_W$ super-quantum-mechanics algebras. Depending on the preserved $SQM$ algebra and the content of the coupled 1d theory, it corresponds either to a vortex or Wilson loop. These Wilson loops reproduce the previous ones and the following provides a consistency check.

We have already noted that on $S_b^3$ we can preserve two distinct $\mathfrak{su}(2|1)\oplus\mathfrak{su}(1|1)$ algebras depending on the enhancement point $m_* = \pm i\frac{b-b^{-1}}{2}$. Inserting a loop operator at the north pole $\theta = 0$ (or south pole $\theta = \pi$) these superalgebras are broken to the $\mathfrak{su}(1|1)\oplus\mathfrak{su}(1|1)$ subalgebras whose Killing vectors are parallel to the loop. These subalgebras are deformed versions of those preserved by a loop on a great circle of the round three-sphere. Consequently we can identify them with the two distinct 1d $\mathcal{N}=4$ super-quantum-mechanics algebras deformed by appropriate background fields. Thus, we can couple $\mathcal{N}=4$ theories on the squashed three sphere to the $\mathcal{N}=4$ super-quantum mechanics on the loops at the poles.

More explicitly, let us go to $\theta = 0$ and the enhancement point $i\frac{b-b^{-1}}{2}$. Then the $\mathfrak{su}(1|1)\oplus\mathfrak{su}(1|1)$ superalgebra preserved by the insertion of the loop can be identified with $SQM_V$, compatible with the discussion in subsection 3.2. Coupling the 3d theory to the same 1d theory as in the round sphere case, but with the deformed or additional background couplings, we get a vortex loop on the squashed three sphere. If we are at the same enhancement point, but at the south pole $\theta = \pi$, the loop preserves the same $\mathfrak{su}(1|1)\oplus\mathfrak{su}(1|1)$. However,

due to the position dependence of the $S_b^3$ Killing spinors (5), it is now mapped to $SQM_W$ and we can couple the correct 1d theory to get a Wilson loop. So we can simultaneously have Wilson loops at $\theta = \pi$ and vortex loops at $\theta = 0$ preserving the same four supercharges.

Now we keep our focus on the vortex loop at $\theta = 0$ and compute its expectation value as we squash the sphere. On the round sphere the vortex loop expectation value takes the form after localization [11]

$$\langle V(\theta = 0)\rangle(b = 1; m, \eta) = \mathcal{N}\, W^{fl} \int d\sigma^{\text{rank }G} \mathcal{F}_{3d}(b = 1; \sigma, m, \eta) \mathcal{I}(b = 1; \sigma, m)\,, \quad (37)$$

where $\mathcal{F}_{3d}$ is the integrand of the localized 3d partition function, $\mathcal{I}$ is the index of the 1d theory we are coupling to it, $\mathcal{N}$ a normalization factor and $W^{fl}$ some flavor Wilson loops.

From localization we see that the vortex loop expectation value on the squashed sphere at the enhancement point is the following generalization of (37):

$$\langle V(\theta = 0)\rangle\left(b; m; \xi; i\frac{b - b^{-1}}{2}\right)$$
$$= \mathcal{N}\, W^{fl} \lim_{z \to b} \int d\hat{\sigma}^{\text{rank }G} \mathcal{F}_{3d}\left(b; \hat{\sigma}; m; \xi; i\frac{b - b^{-1}}{2}\right) \mathcal{I}_{\theta=0}(b; \hat{\sigma}; bm; z)\,. \quad (38)$$

Squashing the sphere the 3d integrand has been replaced by the corresponding squashed sphere expression. We also need the dependence of the 1d index $\mathcal{I}$ on the squashing parameter. The size of the loop is $2\pi r_3 \frac{Q}{2} b$ and thus one would expect that the kinetic piece of modes in the one-loop determinants of the localized index should behave as $(Qb)^{-1}$. However, as we noted already at the end of section 2, we have to remember the couplings to the additional background fields and their effect on indices. Squashing the $S^3$ not only changes the length of the loop but it also introduces additional background fields. To preserve the 4 supercharges, the 1d theory has to couple to these background fields, including the dual gravi-photon from the twist in the compactification from $S^1 \times S^3$. On the circle at $\theta = 0$ this twist acts as a translation proportional to $\frac{(b - b^{-1})}{Q}$. Thus it changes the $b$ dependence of the kinetic piece for modes to $\frac{b}{Q}$. Consequently, we get that the index on the squashed sphere for the 1d theory on the loop at $\theta = 0$ is given in terms of the index on the round sphere as:

$$\mathcal{I}_{\theta=0}(b; \hat{\sigma}; m; z) = \mathcal{I}_{\theta=0}\left(1; \frac{\hat{\sigma}}{b}; \frac{m}{b}; \frac{z}{b}\right)\,. \quad (39)$$

The 3d integrand $\mathcal{F}_{3d}$ on the squashed sphere at the enhancement point $m_* = i\frac{b - b^{-1}}{2}$ can also be related to its round sphere counterpart. Using (13) and (14) we get

$$\mathcal{F}_{3d}\left(b; \hat{\sigma}; m; \xi; i\frac{b - b^{-1}}{2}\right) = b^{-\text{rank }G} \mathcal{F}_{3d}\left(1; \frac{\hat{\sigma}}{b}; \frac{m}{b}; b\xi; 0\right)\,. \quad (40)$$

Inserting the relations (39) and (40) we can rewrite the vortex loop expectation value 38 in terms of the round sphere 1d index and 3d integrand

$$\langle V(\theta = 0)\rangle\left(b; m; \xi; i\frac{b - b^{-1}}{2}\right)$$
$$= \mathcal{N}\, W^{fl} \lim_{z \to 1} \int d\hat{\sigma}^{\text{rank }G} b^{-\text{rank }G} \mathcal{F}_{3d}\left(1; \frac{\hat{\sigma}}{b}; \frac{m}{b}; b\xi; 0\right) \mathcal{I}_{\theta=0}\left(1; \frac{\hat{\sigma}}{b}; \frac{m}{b}; z\right)\,. \quad (41)$$

Rescaling then the Coulomb branch parameter $\frac{\hat{\sigma}}{b} \to \hat{\sigma}$ as well as the mass parameter $\frac{m}{b} \to m$ and the FI coefficient $b\xi \to \xi$, we see that the expectation value of $\frac{2}{3}$-BPS vortex loops becomes

independent of the squashing parameter

$$\langle V(\theta=0)\rangle \left( b; bm; \frac{\xi}{b}; \mathrm{i}\frac{b-b^{-1}}{2}\right) = \langle V(\theta=0)\rangle (1; m; \xi; 0)\,. \tag{42}$$

## 4 Testing dualities

In this section we test mirror dualities on the squashed three-sphere by matching partition functions and loop operator expectation values for dual pairs of theories. Previously this has only been considered for linear quiver theories without Chern-Simons terms [42]. Using enhanced supersymmetry and numerical evaluations we will circumvent the difficulties for analytic tests that arise from the squashing in the localized expressions for partition functions and loop operator expectation values.

### 4.1 Dualities and the enhanced supersymmetry

In the previous section we showed that the squashing dependence drops out of $\frac{2}{3}$-BPS loop operator expectation values. Similarly it was noted in [7] that once six supercharges are preserved on the squashed sphere the localized partition function becomes independent of $b$ assuming the correct $b$ dependence for the mass and the FI parameters. Here we spin this around and use these relations between squashed and round sphere partition functions to lift equalities of mirror dual partition functions to the squashed sphere with six supercharges.

Mirror dualities of D3-D5-NS5 Hanany-Witten brane systems on the round sphere match the partition functions [22],

$$Z_{el}(b=1; \vec{\eta}, \vec{m}; m_*=0) = Z_{mag}(b=1; \vec{m}, \vec{\eta}; m_*=0)\,, \tag{43}$$

where $\vec{\eta}$ designates the FI parameters and $\vec{m}$ the hypermultiplet masses in the "electric" theory. FI parameters and hypermultiplet masses get swapped under the duality. Combining this with our result on squashing independence, we find:

$$
\begin{aligned}
Z_{el}\left( b; \vec{\eta}, \vec{m}; m_*=\mathrm{i}\frac{b-b^{-1}}{2}\right) &= Z_{el}\left(1; b\vec{\eta}, b^{-1}\vec{m}; m_*=0\right) \\
&= Z_{mag}\left(1; b^{-1}\vec{m}, b\vec{\eta}; m_*=0\right) \\
&= Z_{mag}\left( b; \vec{m}, \vec{\eta}; m_*=-\mathrm{i}\frac{b-b^{-1}}{2}\right).
\end{aligned}
\tag{44}
$$

Thus, even on the squashed sphere with six supercharges, when matching the partition functions the FI parameters and masses get swapped with no additional factors of the squashing parameter. Note that the sign flip for the special mass $m_*$ is in accordance with the exchange of Higgs and Coulomb branches and the special mass $m_*$ gauging the axial $U(1)_{ax}$ of $SU(2)_C \times SU(2)_H$.

Less trivial is the mapping for the duality of ABJM and $\mathcal{N}=8$ SYM. Here one finds

$$
\begin{aligned}
Z_{\mathrm{ABJM}}\left( b; \eta, \mu; m_*=\mathrm{i}\frac{b-b^{-1}}{2}\right) &= Z_{\mathrm{ABJM}}\left(1; \frac{Q}{2}\eta - \frac{b-b^{-1}}{8}\mu, \frac{Q}{2}\mu - 2(b-b^{-1})\eta; 0\right) \\
&= Z_{\mathcal{N}=8}\left(1; b^{-1}\left(\frac{\mu}{2}+2\eta\right), b\left(\frac{\mu}{2}-2\eta\right); 0\right) \\
&= Z_{\mathcal{N}=8}\left( b; \frac{\mu}{2}+2\eta, \frac{\mu}{2}-2\eta; m_*=-\mathrm{i}\frac{b-b^{-1}}{2}\right),
\end{aligned}
\tag{45}
$$

where $\eta$ is the FI parameter and $\mu$ is a hypermultiplet mass term in ABJM.[7] The second step comes from the round sphere mapping [22]. Again we notice that at the SUSY enhancement point the round sphere map of the parameters simply lifts to the squashed sphere, with $m_*$ flipping sign.

Let us now include line operators starting with the abelian case. The Wilson loop for an abelian group maps under mirror symmetry to the vortex loop for the topological $U(1)_J$ symmetry. The matching of their expectation values under mirror symmetry has been checked on the round sphere [15, 16]. Due to the simplicity of the abelian lines, we can hide many factors into the charge $q$ of the loop operator. The round sphere duality lifts to the squashed sphere in the form

$$\langle W = e^{2\pi q\hat{\sigma}}\rangle\left(b; \vec{\eta}, \vec{m}; m_* = -i\frac{b - b^{-1}}{2}\right) = \langle W = e^{2\pi\frac{q}{b}\hat{\sigma}}\rangle\left(1; \frac{\vec{\eta}}{b}, b\vec{m}; m_* = 0\right)$$

$$= Z_{dual}\left(b = 1; FI = b\vec{m}, \text{mass} = \frac{\vec{\eta}}{b} + i\frac{q}{b}; m_* = 0\right)$$

$$= \langle V\rangle\left(b; k = 0; q; \vec{m}, \vec{\eta}; m_* = i\frac{b - b^{-1}}{2}\right). \tag{46}$$

The notation in the first line means the expectation value of a Wilson loop corresponding to that specific insertion into the matrix model. For completeness we carry along possible FI parameters $\vec{\eta}$ and hypermultiplet masses $\vec{m}$. The shift in the second line only affects the component of $\vec{\eta}$ for the same group as the Wilson line is charged under. In the last line we see that we get a vortex loop of charge $q$ as the dual of the Wilson loop with the same charge, giving us once again the same map of loop operators as on the round sphere. Note that this computation holds for *all abelian closed $\frac{1}{3}$-BPS loops*.

For the generic non-abelian loop operators we can lift the duality map for $\frac{1}{2}$-BPS loops discussed in [11] to the squashed sphere and its $\frac{2}{3}$-BPS loops. We compute along the same lines as before:

$$\langle W_{\mathcal{R}}(\theta = 0)\rangle\left(b; \xi, m; m_* = -i\frac{b - b^{-1}}{2}\right) = \langle W_{\mathcal{R}}(\theta = 0)\rangle\left(1; \frac{\xi}{b}, bm; m_* = 0\right)$$

$$= \langle V(\theta = 0)\rangle\left(1; bm, \frac{\xi}{b}; m_* = 0\right)$$

$$= \langle V(\theta = 0)\rangle\left(b; m, \xi; m_* = i\frac{b - b^{-1}}{2}\right). \tag{47}$$

We have used the squashing independence of the Wilson loop expectation value, the round sphere duality and the squashing independence of the vortex loop expectation value. The conclusion is that mirror symmetry maps Wilson loops in a theory at one of the enhancement points to vortex loops in the mirror theory at the other enhancement point.

The lift from the round sphere is compatible with the mirror symmetry results of [41, 42]. Recursively applying Aharony duality they reduce the rank of the mirror duality down to a base case that can be proven analytically. Aharony duality maps each SUSY enhancement point to itself. Following the recursive algorithm of [42] at the enhancement point we then see that the base case gives the one single sign flip in the special mass consistent with mirror symmetry.

## 4.2 Numerical results

In this subsection we apply numerics to test the duality between ABJM and $\mathcal{N} = 8$ SYM on the squashed sphere with only four preserved supercharges. We compute the partition functions

---

[7]In the notation of [9] we have exchanged the mass $m_1$ for the FI parameter $\frac{\eta}{2}$, the mass $m_2$ is $\mu$ and $m_3$ is the special mass $m_*$. Compared to [22] we have renamed the parameters $\zeta$ to $\eta$ and $\xi$ to $\mu$.

at $m_* = 0$ on both sides of the duality at low rank and check that they agree within numerical errors. We present most results in terms of the free energy $F = -\ln Z$ as it is less prone to numerical instabilities.

The double sine function which appears in the localization integral has an exponential fall-off for large argument. Thus, a simple way to numerically approximate the value of the partition function is to sum the values of the integrand over a finite size evenly spaced grid. The downfall of this elementary approach is that the number of points sampled grows exponentially with the dimension of the integral and thus it limits us to the lowest ranks of the gauge group. For higher ranks better algorithms like Monte-Carlo methods should be used to keep the computational power manageable.

Using the outlined approach we evaluate the partition functions for $U(N)_1 \times U(N)_{-1}$ ABJM and $U(N) \mathcal{N} = 8$ SYM for $N = 2$ and 3. The complete expressions are given in Appendix B and the numerical evaluation of the double sine function is described in Appendix A. For $N = 2$ we compute the numerical estimates for the range $[1, 10]$ of the squashing parameter as well as two additional deformation parameters ranging in $[0, 0.89]$ and $[0, 2.98]$ and corresponding to the FI coefficient and bifundamental mass in the ABJM theory. For $N = 3$ we varied the squashing parameter in the range $[1, 10]$ but kept all other parameters zero.

Before presenting the results let us discuss the expected numerical precision. The evaluation of the double sine function contains a numerical integration which was performed using *scipy*'s quadrature library with absolute and relative error tolerances set to $2 \cdot 10^{-7}$. In parameter regions where the oscillatory behavior of the integrand has little effect, this accounts well for the differences observed, this can be seen in parts of Figures 3, 4 and 5. We are thus confident that the finiteness and discretization of the domain do not contribute significantly to errors. For most of the parameter region considered cancellations are however important and it is well known that this reduces the numerical precision by several orders of magnitude. To estimate the precision lost through the oscillatory behavior of the integrands we compare against analytic results. On the round and $b^2 = 3$ squashed spheres the exact values of the partition function have been computed for zero mass and zero FI parameter [49, 50]. In Table 1 we compare these results to the numerical free energy and find a agreement with relative errors of order $10^{-5}$ for $N = 2$ and $10^{-3}$ for $N = 3$. This gives an approximate value for the numerical accuracy of our algorithm at zero mass and FI parameter. Notably the numerical accuracy appears to not depend much on the squashing parameter. For $N = 2$, [44, 45] computed the round sphere partition function of ABJM analytically. In Figure 1 we show the relative difference of this result to our numerical values. This gives an estimate for the accuracy of our numerical evaluations and we thus expect our results for the partition functions to match within a relative difference of between $10^{-5}$ to $10^{-2}$. As we will discuss later, areas where the error reaches the upper bound $10^{-2}$ of this estimate surround zeros of the partition function and thus a loss of numerical precision is expected.

At rank $N = 2$ the results of the numerical evaluations of the real parts of the free energy $\mathrm{Re}(F) = -\ln|Z|$ are plotted[8] in Figures 2–5 in the $b = 1$, $b = 4$, $\mu = 0$ and $\eta = 0$ planes respectively and in Figure 6a at fixed $\mu, \eta$. Below each plot we also display the absolute value of the relative difference $\frac{|\mathrm{Re}(F)_{ABJM} - \mathrm{Re}(F)_{SYM}|}{|\mathrm{Re}(F)_{SYM}|}$ to show how well the functions match. Comparing against Figure 1 we see that the numerical values of the two partition functions agree at least as good as expected for all of the parameter values displayed. This is strong evidence that the two partition functions are equal for the range of parameters considered. The figures presented in this paper show a selection of all data computed. In [51] we show in videos how the free energy evolves as the sphere is squashed, extending on Figures 2 and 3 by showing

---

[8]As the ABJM and $\mathcal{N} = 8$ SYM free energies agree very well on the whole range of parameters considered, the *python* library *Matplotlib* used to create the graphs chooses to only draw one of the functions for most of the parameter ranges.

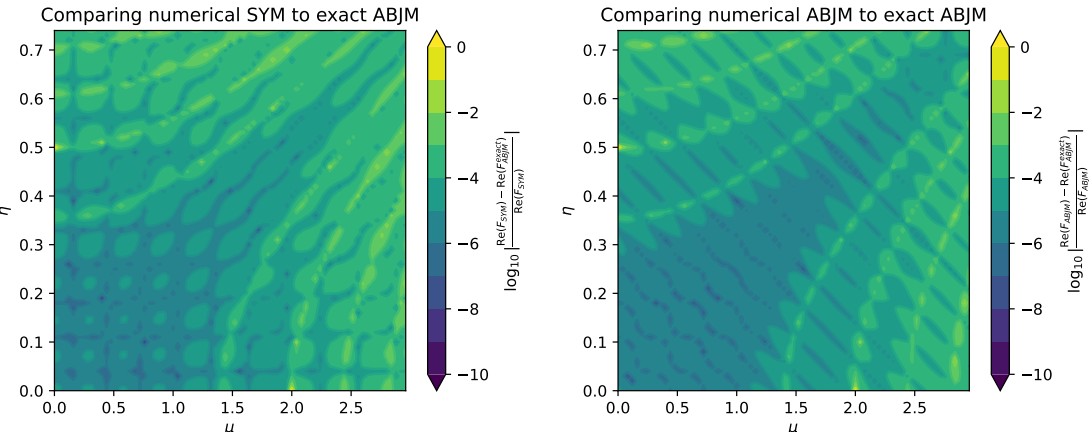

Figure 1: Comparison of the round sphere numerical results to the exact analytical value of the free energy. These plots set the scale for the accuracy we can expect for the numerical evaluations of the free energy at all values of the squashing, mass and FI parameters.

1001 values of the squashing parameter in the range $[1,6]$.

Most notable in the Figures 2–5 are the sharp ridges that appear on the surfaces when the FI or mass parameter is large enough. In Figure 6a this manifests itself as an outlier point for the $2\eta = \mu = 0.98$ curve. In Figure 6d we zoom in on the range of squashing parameters around this one outlier. We see that it corresponds to a very sharp kink in the free energy. In Figure 6e we look at the real and imaginary parts of the partition function and see that these kinks and ridges correspond to zeros of the partition function. Thus in the three dimensional $(b, \eta, m_{bif})$ parameter space the partition function has two dimensional surfaces of zeros. On the round sphere these Lee-Yang zeros were found from the analytic expression for the $U(2)_1 \times U(2)_{-1}$ ABJM partition function [44, 45]. Translating conventions, the zeros are distributed at $b = 1$ along the hyperbolas

$$\mu^2 - 4\eta^2 = 2n, \qquad n \in \mathbb{Z} \setminus \{0\}. \tag{48}$$

In Figure 7 we see that this matches perfectly with our numerics. Increasing $b$ the zeros move to larger values of FI and mass parameters. While the phases responsible for the zeros pose a challenge to the numerical evaluations and the error estimates grow in the proximity of

Table 1: Comparing the exact free energy to the numerical ABJM and $\mathcal{N} = 8$ values and the perturbative large $N$ conjecture for $b^2 \in \{1, 3\}$ at ranks $N$ two and three. The relative difference of order $10^{-5}$, resp. $10^{-3}$, shows that the numerical approximation is good.

| $N$ | $b^2$ | Exact | $\mathcal{N} = 8$ SYM | ABJM | large $N$ |
|-----|-------|-------|-----------------------|------|-----------|
| 2 | 1 | $-\log\left(\dfrac{1}{16\pi}\right) \approx 3.917319$ | 3.917361 | 3.917371 | 3.917307 |
| | 3 | $-\log\left(\dfrac{1}{12\sqrt{3}\pi} - \dfrac{1}{81}\right) \approx 5.819526$ | 5.819535 | 5.819539 | 5.830818 |
| 3 | 1 | $-\log\left(\dfrac{-3 + \pi}{64\pi}\right) \approx 7.25841$ | 7.26001 | 7.26613 | 7.25840 |
| | 3 | $-\log\left(\dfrac{5}{2187} - \dfrac{1}{72\pi^2} - \dfrac{1}{216\sqrt{3}\pi}\right) \approx 10.4768$ | 10.4779 | 10.4770 | 10.4881 |

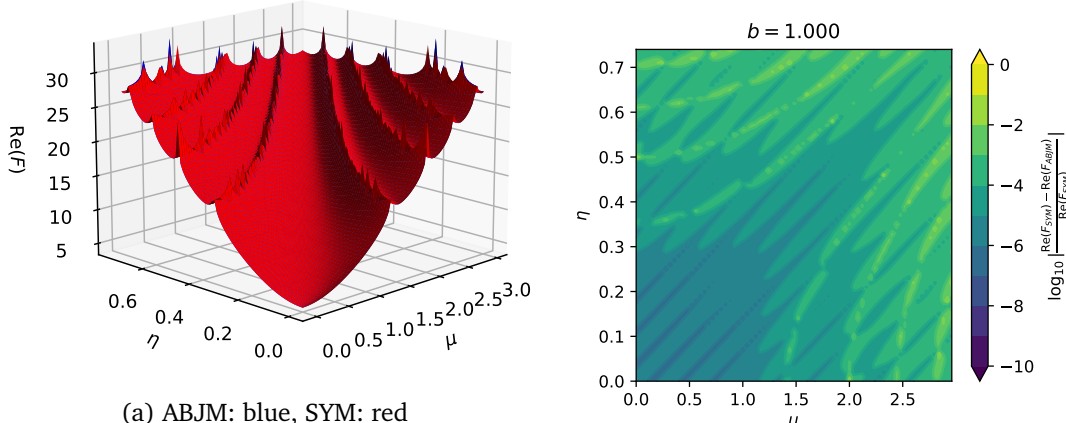

(a) ABJM: blue, SYM: red

Figure 2: We plot the real parts of the ABJM and $\mathcal{N} = 8$ $U(2)$ free energies against the bi-fundamental mass $\mu$ and the FI parameter $\eta$ at fixed squashing parameter $b = 1$. We observe that the graphs have some ridges along certain curves in the $(\eta, \mu)$ plane and errors grow along them due to numerical instability.

the zeros, the fact that the locations of the zeros agree for ABJM and $\mathcal{N} = 8$ SYM partition functions signals that the two functions are indeed equal.

In Figure 8 we plot the real parts of the numerical values of the ABJM and SYM free energies and their relative difference for rank three. Agreement is roughly within the $10^{-3}$ that we expect from the comparison to the analytic result on the round sphere and at $b^2 = 3$.

Based on these numerical results we feel confident to conjecture that the following two partition functions are equal for all values of $b, \eta, \mu$ and $m_*$

$$
Z_{\text{ABJM}}(\mu, \eta) = \frac{1}{(N!)^2} \int d^N \sigma d^N \widetilde{\sigma} e^{i\pi \sum_i (\sigma_i^2 - \widetilde{\sigma}_i^2 + 2\eta(\sigma_i + \widetilde{\sigma}_i))} \prod_{i<j} 16 \sinh\left(\pi b^{\pm 1} \sigma_{ij}\right) \sinh\left(\pi b^{\pm 1} \widetilde{\sigma}_{ij}\right)
$$
$$
\times \prod_{i,j} \frac{1}{S_2\left(\frac{Q}{4} + i\left(\sigma_i - \widetilde{\sigma}_j \pm \frac{\mu + m_*}{2}\right)\right) S_2\left(\frac{Q}{4} + i\left(-\sigma_i + \widetilde{\sigma}_j \pm \frac{\mu - m_*}{2}\right)\right)}, \quad (49)
$$

$$
Z_{SYM} = \frac{1}{N!} \int d\sigma_i e^{2\pi i (\frac{\mu}{2} + 2\eta) \sum_i \sigma_i} \prod_{i<j} \frac{4 \sinh\left(\pi b^{\pm 1} \sigma_{ij}\right)}{S_2\left(\frac{Q}{2} + i m_* \pm i\sigma_{ij}\right)}
$$
$$
\times \frac{1}{\prod_i S_2\left(\frac{Q}{4} + \frac{i m_*}{2} \pm i\sigma_i\right) \prod_{i,j} S_2\left(\frac{Q}{4} + \frac{i m_*}{2} \pm i(\sigma_{ij} + \frac{\mu}{2} - 2\eta)\right)}. \quad (50)
$$

Note how the signs for $m_*$ in the super-Yang-Mills partition function are opposite compared to (13), (14) as required by the exchange of the Higgs and the Coulomb branches under mirror symmetry. We expect that the techniques of [42, 43] can be applied to prove this equality.

Included in Table 1 and Figures 6a, 8a is also the all order large $N$ perturbative free energy of ABJM as it was conjectured in [46] (cf. Appendix C for the details). In Figure 6c we observe that at rank $N = 2$ the relative difference between the numerical and the perturbative free energy is significantly larger than the estimated numerical error. This difference measures the non-perturbative contributions to the ABJM free energy coming from worldsheet and membrane instantons. At rank $N = 3$ the difference is not quite as clearcut because errors

$b = 4.000$

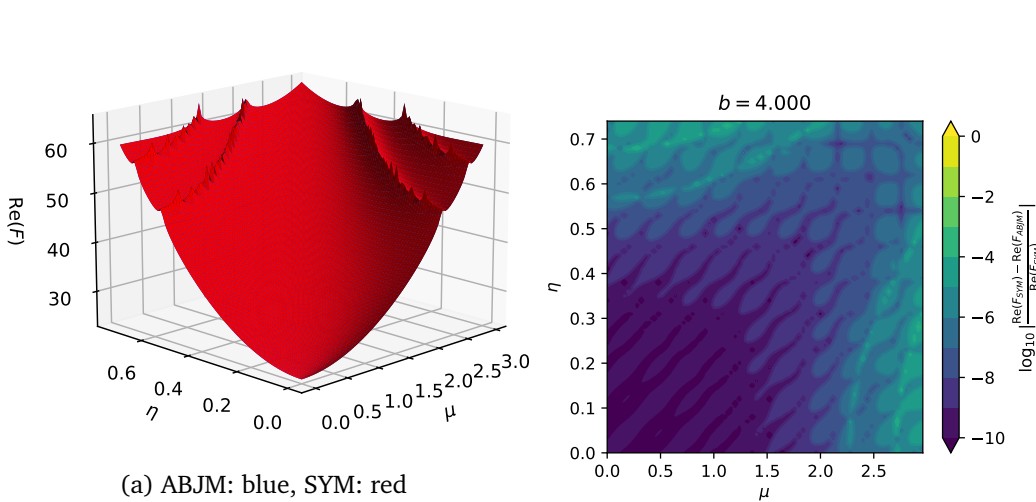

(a) ABJM: blue, SYM: red

Figure 3: We plot the real parts of the ABJM and $\mathcal{N} = 8$ $U(2)$ free energies against the bi-fundamental mass $\mu$ and the FI parameter $\eta$ at fixed squashing parameter $b = 4$. We observe that the graphs have some ridges/kinks along certain curves in the $(\eta, \mu)$ plane and errors grow along them due to numerical instability.

$\mu = 0.00$

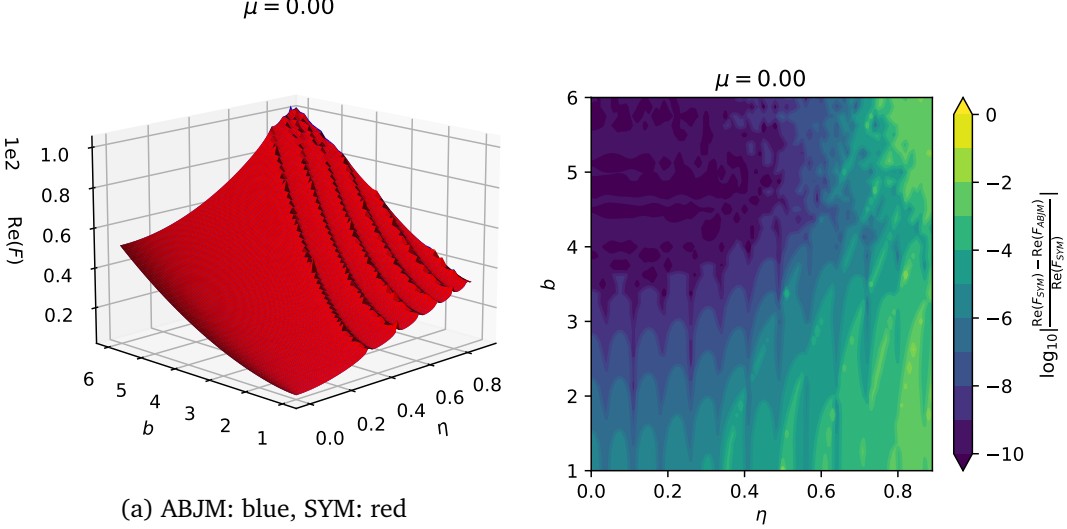

(a) ABJM: blue, SYM: red

Figure 4: We plot the real parts of the ABJM and $\mathcal{N} = 8$ $U(2)$ free energies against the squashing parameter $b$ and the FI parameter $\eta$ at fixed bi-fundamental mass $\mu = 0$. We observe that the graphs have some ridges along certain curves in the $(b, \eta)$ plane and errors grow along them due to numerical instability.

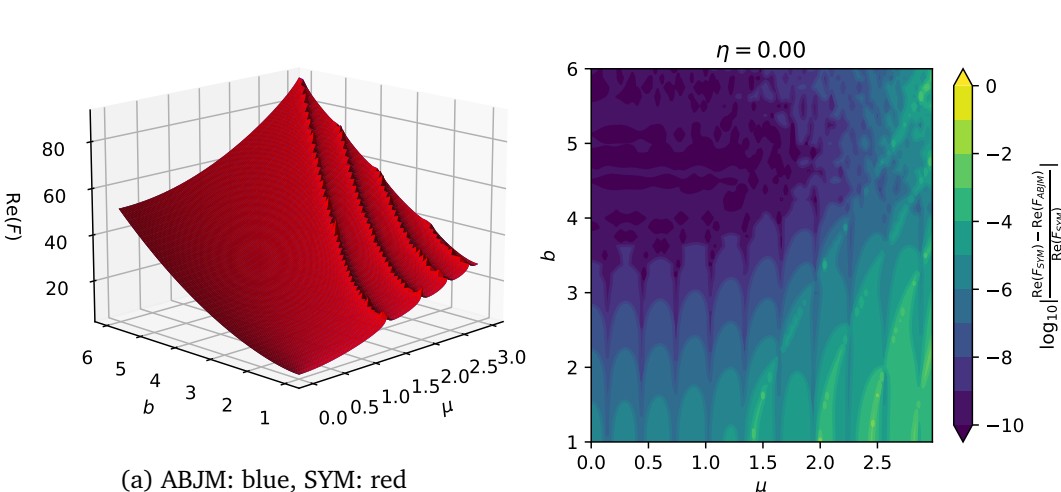

(a) ABJM: blue, SYM: red

Figure 5: We plot the real parts of the ABJM and $\mathcal{N} = 8$ $U(2)$ free energies against the squashing parameter $b$ and the bi-fundamental mass $\mu$ at fixed FI parameter $\eta = 0$. We observe that the graphs have some ridges along certain curves in the $(b, \mu)$ plane and errors grow along them due to numerical instability.

in our approximation of the ABJM free energy at this rank are larger. However the relative difference in Figure 8c is still much larger than our error estimate on the numerics, meaning we confidently observe non-perturbative contributions to the free energy. Furthermore, for ranks two and three the overall behavior is similar. Starting from the round sphere the instanton contributions are negative until $b \approx 5$ where they start growing and become positive at $b \approx 9$. This suggests a general pattern and it would be interesting to understand why the non-perturbative contributions behave this way. However, it should be noted that the $N$ independent factor (C.5) to the conjectured perturbative partition function has not yet been thoroughly tested. Thus we might be seeing corrections to this factor instead of observing the non-perturbative corrections to the partition function.

## Acknowledgments

We are greatful to U. Naseer and J. A. Minahan for their help throughout many stages of this project, the discussions, the valuable suggestions and their comments on the draft. We also thank N. Bobev, J. Hong and V. Reys for their careful reading of the draft and L. Cassia and M. Zabzine for valuable discussions.

**Funding information** This research was supported in part by Vetenskapsrådet under grants #2016-03503 and #2020-03339.

## A The double sine function

We use the definition of [52] for the double sine function:

$$S_2(x; \omega_1, \omega_2) = \prod_{m,n \geq 0} \frac{m\omega_1 + n\omega_2 + x}{(m+1)\omega_1 + (n+1)\omega_2 - x}, \tag{A.1}$$

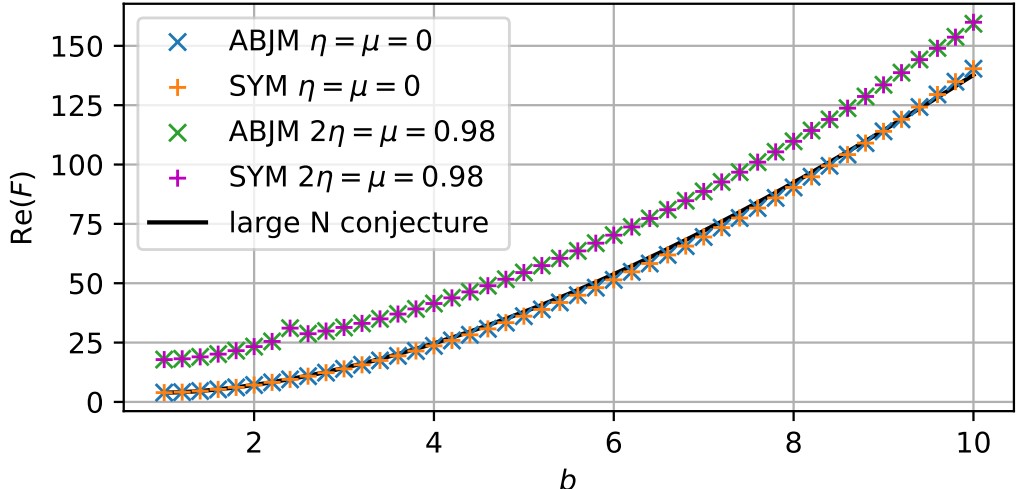

(a) Real parts of the free energies for two pairs of values of the deformation parameters. Note how the point at $b = 2.4$ and $2\eta = \mu = 0.98$ is offset. We have also included the perturbative large $N$ conjecture for $\eta = \mu = 0$.

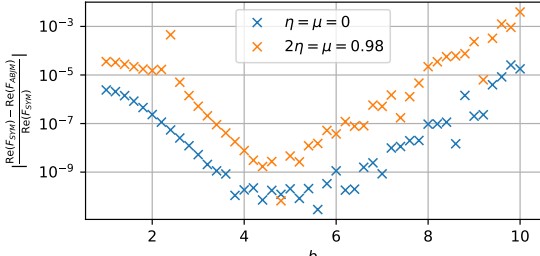

(b) Relative difference of the numerical approximations to the free energy. Note that close to the kink the error grows.

(c) Relative difference of the computed free energy to the all order large $N$ conjecture.

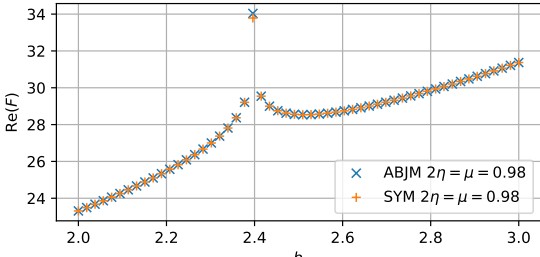

(d) Zooming in around $b = 2.4$, $2\eta = \mu = 0.98$ to highlight how pronounced the kink in the graph is.

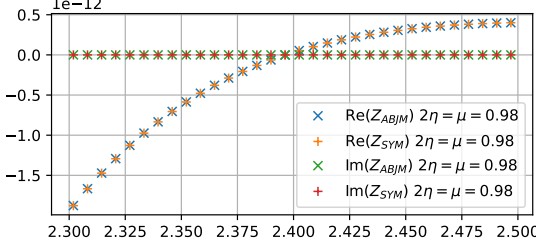

(e) The kink in the real part of the free energy corresponds to a zero of the partition function.

Figure 6: Numerical values of the ABJM and $\mathcal{N} = 8$ free energies for $U(2)$ gauge groups plotted against the squashing parameter $b$ for $(\eta, \mu) \in \{(0, 0), (0.49, 0.98)\}$. For $\eta = \mu = 0$ we compare against the all-order large $N$ conjecture for the ABJM free energy. We show that kinks in the curves correspond to zeros of the partition function.

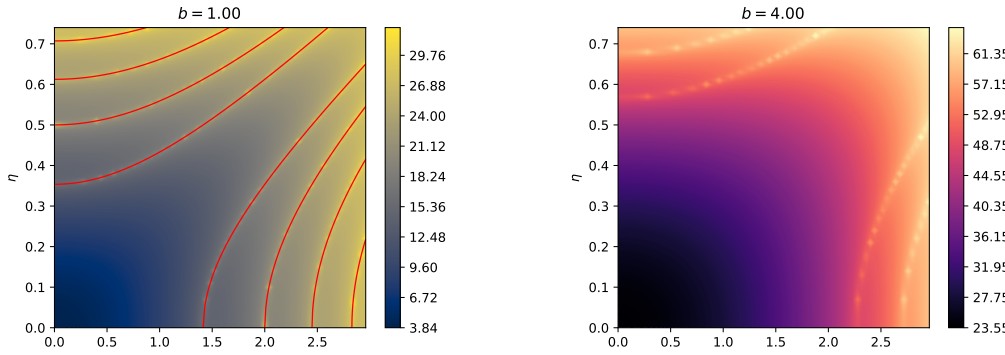

Figure 7: For two values of $b$ we plot the real part of the free energy of the $\mathcal{N} = 8$ super-Yang-Mills as a filled contour plot. Note how the ridges from the 3d graphs are visible. For $b = 1$ we also plot in red the locations of the zeros as found by [45] in the ABJM theory and we note that these hyperbolas visually fit very well to the locations of the ridges.

where the infinite product has to be zeta-function regularized. It relates to the definition $s_b$ used in most physics literature [4] and the hyperbolic gamma function of [38] as

$$s_b(x) = S_2\left(x + i\frac{Q}{2}; ib, ib^{-1}\right) = \frac{1}{\Gamma_h(x + i\frac{Q}{2}; ib, ib^{-1})}, \tag{A.2}$$

with $Q = b + \frac{1}{b}$. For numerical evaluation it is advantageous to rewrite the double sine function in terms of an integral [53]

$$S_2(x; \omega_1, \omega_2) = \exp\left(\frac{\pi i}{2} B_{22}(x; \omega_1, \omega_2) + \int_{\mathbb{R}+i0} \frac{e^{xz}}{(e^{\omega_1 z} - 1)(e^{\omega_2 z} - 1)} \frac{dz}{z}\right), \tag{A.3}$$

where the contour avoids zero by going into the upper half-plane. The Bernoulli polynomial $B_{22}$ is defined as

$$B_{22}(x; \omega_1, \omega_2) = \frac{z^2}{\omega_1 \omega_2} - \frac{\omega_1 + \omega_2}{\omega_1 \omega_2} z + \frac{\omega_1^2 + \omega_2^2 + 3\omega_1 \omega_2}{6\omega_1 \omega_2}. \tag{A.4}$$

The integral representation for the double sine can be evaluated numerically with good precision. Implementing the contour by shifting the integration variable by a small imaginary number, it can be computed using *scipy.integrate.quad* in *python* with finite cutoffs on the integration interval.

# B Partition function for ABJM and its mirror at low rank

In this appendix we write down the expressions for the partition functions and simplify them to the expressions that we numerically evaluate in subsection 4.2.

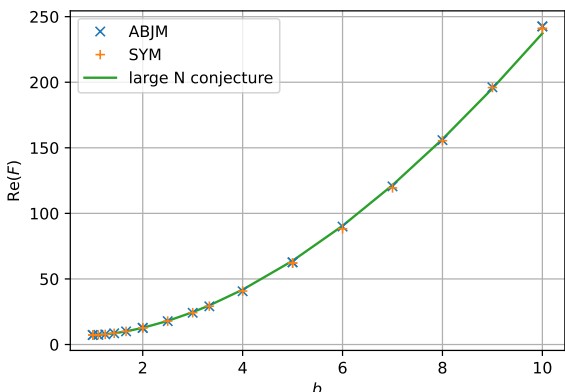

(a) Real part of the computed ABJM and $\mathcal{N} = 8$ SYM free energies at rank three and the perturbative all order large $N$ conjecture.

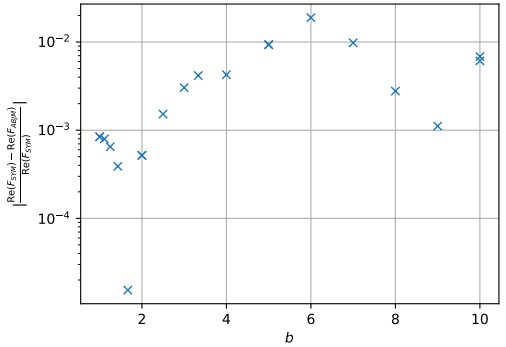

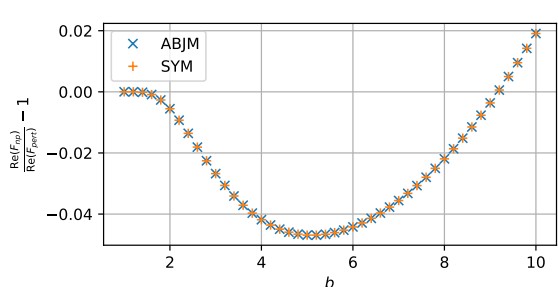

(b) Relative difference of the numerical free energies.

(c) Relative difference of the computed free energies to the conjectured perturbative all order large $N$ expression for the free energy.

Figure 8: The numerical results for the free energies of ABJM and $\mathcal{N} = 8$ SYM for $U(3)$ gauge groups plotted against the squashing parameter $b$. We compare the two computed partition functions to each other and to the conjectured perturbative large $N$ expression.

## B.1 Rank 2

First we look at the $U(2)_1 \times U(2)_{-1}$ ABJM theory which is mirror dual to $U(2)$ SYM with an adjoint and a fundamental hypermultiplet. For general $U(N)_1 \times U(N)_{-1}$ gauge group we have

$$
Z_{\mathrm{ABJM}}(\mu, \eta) = \frac{1}{(N!)^2} \int d^N \sigma \, d^N \widetilde{\sigma} \exp\left( i\pi \sum_i \sigma_i^2 - \widetilde{\sigma}_i^2 + 2\eta (\sigma_i + \widetilde{\sigma}_i) \right)
$$

$$
\times \prod_{i<j} 16 \sinh\left(\pi b^{\pm 1} \sigma_{ij}\right) \sinh\left(\pi b^{\pm 1} \widetilde{\sigma}_{ij}\right) \prod_{i,j} \frac{1}{S_2\left(\frac{Q}{4} \pm i\frac{\mu}{2} \pm i\left(\sigma_i - \widetilde{\sigma}_j\right)\right)}, \quad \text{(B.1)}
$$

in terms of the FI parameter $\eta$ and the mass $\mu$ for the bi-fundamentals. We use the notation $S_2(u \pm v) = S_2(u+v)S_2(u-v)$ and we suppress the squashing parameter $b$ in the double sine functions. Specializing to $N = 2$, out of the four integration variables only three appear in the one-loop determinants

$$
x = \sigma_1 - \sigma_2, \qquad y = \sigma_1 - \widetilde{\sigma}_1, \qquad z = \sigma_2 - \widetilde{\sigma}_2. \quad \text{(B.2)}
$$

After this change of variables, the $\sigma_2$ integral gives a Dirac-$\delta$-function which in turn allows us to perform the $z$ integral. Further simplification comes from shifting variables $x \to x + y$, $y \to y - 2\eta$. In the end we get

$$Z_{\text{ABJM}}(\mu, \eta) = 4 \int dx\,dy\, e^{2\pi i x y} \sinh\big(\pi b^{\pm 1}(x+y)\big) \sinh\big(\pi b^{\pm 1}(x-y)\big)$$
$$\times \frac{1}{S_2\big(\frac{Q}{4} \pm i\frac{\mu}{2} \pm 2i\eta \pm iy\big) S_2\big(\frac{Q}{4} \pm i\frac{\mu}{2} \pm 2i\eta \pm ix\big)}. \quad \text{(B.3)}$$

At generic $N$ the partition function of its mirror dual is given by

$$Z = \frac{1}{N!} \int d^N \sigma\, e^{2\pi i \eta_m \sum_i \sigma_i} \frac{\prod_{i<j} 4 \sinh\big(\pi b^{\pm 1} \sigma_{ij}\big)}{\prod_i S_2(\frac{Q}{4} \pm i\sigma) \prod_{i,j} S_2(\frac{Q}{4} \pm (\sigma_{ij} + \frac{\mu}{2}))}. \quad \text{(B.4)}$$

Specializing to $N = 2$ this simplifies to

$$Z = \frac{1}{2S_2^2\big(\frac{Q}{4} \pm i\frac{\mu_m}{2}\big)} \int d\sigma_1 d\sigma_2\, e^{2\pi i \eta_m(\sigma_1 + \sigma_2)} 4 \sinh\big(\pi b^{\pm 1} \sigma_{12}\big)$$
$$\times \frac{1}{S_2\big(\frac{Q}{4} \pm i\sigma_1\big) S_2\big(\frac{Q}{4} \pm i\sigma_2\big) S_2\big(\frac{Q}{4} \pm i\sigma_{12} \pm i\frac{\mu_m}{2}\big)}. \quad \text{(B.5)}$$

To simplify it, we change variables to $y = \sigma_1 - \sigma_2$ then

$$Z = \frac{1}{2S_2^2\big(\frac{Q}{4} \pm i\frac{\mu_m}{2}\big)} \int d\sigma_1 dy\, e^{2\pi i \eta_m(2\sigma_1 - y)} 4 \sinh\big(\pi b^{\pm 1} y\big)$$
$$\times \frac{1}{S_2\big(\frac{Q}{4} \pm i\sigma_1\big) S_2\big(\frac{Q}{4} \pm i(\sigma_1 - y)\big) S_2\big(\frac{Q}{4} \pm iy \pm i\frac{\mu_m}{2}\big)}. \quad \text{(B.6)}$$

The second term on the second line can be rewritten in terms of its own Fourier transform

$$Z = \frac{1}{2S_2^2\big(\frac{Q}{4} \pm i\frac{\mu_m}{2}\big)} \int dx\,d\sigma_1 dy\, e^{2\pi i \eta_m(2\sigma_1 - y)} 4 \sinh\big(\pi b^{\pm 1} y\big)$$
$$\times \frac{e^{-2\pi i \sigma_1 x} e^{2\pi i y x}}{S_2\big(\frac{Q}{4} \pm i\sigma_1\big) S_2\big(\frac{Q}{4} \pm ix\big) S_2\big(\frac{Q}{4} \pm iy \pm i\frac{\mu_m}{2}\big)}. \quad \text{(B.7)}$$

The $\sigma_1$-integral can be performed exactly and shifting $x \to x + \eta_m$ we find

$$Z_{\text{ABJM,mirror}} = \frac{1}{2S_2^2\big(\frac{Q}{4} \pm i\frac{\mu_m}{2}\big)} \int dx\,dy\, e^{2\pi i y x} 4 \sinh(\pi b y) \sinh\big(\pi b^{-1} y\big)$$
$$\times \frac{1}{S_2\big(\frac{Q}{4} \pm i\eta_m \pm ix\big) S_2\big(\frac{Q}{4} \pm i\frac{\mu_m}{2} \pm iy\big)}. \quad \text{(B.8)}$$

The FI parameters and the masses on both sides of the duality are related as

$$\eta_m = \frac{\mu}{2} + 2\eta, \qquad \mu_m = \mu - 4\eta. \quad \text{(B.9)}$$

## B.2 Rank 3

At rank three we compute the partition functions for zero mass and the FI parameter as the ABJM computation is too unstable to observe the relevant features.

We specialize (B.4) to $N = 3$, set $\mu_m = \eta_m = 0$ and we use the variables $x = \sigma_{12}$, $y = \sigma_{13}$:

$$Z_{SYM} = \frac{4^3}{3!S_2\left(\frac{Q}{4}\right)^6} \int d\sigma_1 dx dy \, \sinh\left(\pi b^{\pm 1} x\right) \sinh\left(\pi b^{\pm 1} y\right) \sinh\left(\pi b^{\pm 1}(x-y)\right)$$
$$\times \frac{1}{S_2\left(\frac{Q}{4} \pm ix\right)^2 S_2\left(\frac{Q}{4} \pm iy\right)^2 S_2\left(\frac{Q}{4} \pm i(x-y)\right)^2}$$
$$\times \frac{1}{S_2\left(\frac{Q}{4} \pm i(x-\sigma_1)\right) S_2\left(\frac{Q}{4} \pm i\sigma_1\right) S_2\left(\frac{Q}{4} \pm i(y-\sigma_1)\right)} . \tag{B.10}$$

For the ABJM, starting from (B.1) we use the variables $U = \sigma_{21}$, $V = \sigma_{31}$, $X = \tilde{\sigma}_1 - \sigma_1$, $Y = \tilde{\sigma}_2 - \sigma_2$, $X = \tilde{\sigma}_3 - \sigma_3$. Then the $\sigma_1$ integral can be performed, giving a $\delta$-function that allows us to also do the $X$ integral. Therefore the integral that we actually evaluate numerically is

$$Z_{ABJM} = \frac{4^6}{(3!)^2} \int dU dV dY dZ \, \exp\left(-2\pi i(UY + VZ + Y^2 + Z^2 + YZ)\right)$$
$$\times \sinh\left(\pi b^{\pm 1} U\right) \sinh\left(\pi b^{\pm 1} V\right) \sinh\left(\pi b^{\pm 1}(U-V)\right)$$
$$\times \sinh\left(\pi b^{\pm 1}(U+2Y+Z)\right) \sinh\left(\pi b^{\pm 1}(V+Y+2Z)\right) \sinh\left(\pi b^{\pm 1}(U-V+Y-Z)\right)$$
$$\times \frac{1}{S_2\left(\frac{Q}{4} \pm i(Y+Z)\right)^2 S_2\left(\frac{Q}{4} \pm iY\right)^2 S_2\left(\frac{Q}{4} \pm iZ\right)^2 S_2\left(\frac{Q}{4} \pm i(U+Y)\right)^2}$$
$$\times \frac{1}{S_2\left(\frac{Q}{4} \pm i(V+Z)\right)^2 S_2\left(\frac{Q}{4} \pm i(U+Y+Z)\right)^2 S_2\left(\frac{Q}{4} \pm i(V+Y+Z)\right)^2}$$
$$\times \frac{1}{S_2\left(\frac{Q}{4} \pm i(U-V-Z)\right)^2 S_2\left(\frac{Q}{4} \pm i(U-V+Y)\right)^2} . \tag{B.11}$$

# C  The all order large $N$ conjecture

Based on exact formulas for the round sphere all order large $N$ partition function [54–56], it has been conjectured and tested in some regimes that the perturbative partition function for the ABJM evaluates to [46]

$$Z(N;k;b) = C(k;b)^{-\frac{1}{3}} e^{\mathcal{A}(k;b)} \text{Ai}(z(N;k;b)), \tag{C.1}$$

in terms of the Airy-function $\text{Ai}(z)$ and the definitions:

$$z(N;k;b) = C(k;b)^{-\frac{1}{3}}(N - B(k;b)), \tag{C.2}$$

$$C(k;b) = \frac{\gamma(b)}{k}, \qquad B(k;b) = \frac{\alpha(b)}{k} + \beta(b)k, \tag{C.3}$$

$$\gamma(b) = \frac{32}{\pi^2\left(b+\frac{1}{b}\right)^4}, \qquad \alpha(b) = -\frac{2}{3}\left(b^2 - 4 + \frac{1}{b^2}\right)\left(b+\frac{1}{b}\right)^{-2}, \qquad \beta(b) = \frac{1}{24}, \tag{C.4}$$

$$\mathcal{A}(k;b) = \frac{1}{4}\left(\mathcal{A}(k(1+ib_+)) + \mathcal{A}(k(1-ib_+)) + \mathcal{A}(k(1+ib_-)) + \mathcal{A}(k(1-ib_-))\right), \qquad (C.5)$$

$$\mathcal{A}(k) = \frac{2\zeta(3)}{\pi^2 k}\left(1 - \frac{k^3}{16}\right) + \frac{k^2}{\pi^2}\int_0^\infty dx\,\frac{x}{e^{kx}-1}\log\left(1-e^{-2x}\right), \qquad (C.6)$$

$$b_\pm = \frac{1}{2\sqrt{2}}\left(b - \frac{1}{b}\right)\sqrt{b^2 + \frac{1}{b^2} \pm \sqrt{b^4 + 14 + \frac{1}{b^4}}}\,. \qquad (C.7)$$

We evaluate this in *python* using the implementation of the Airy function in *scipy.special* and approximating $\mathcal{A}$ by an integration on a finite interval using *scipy.integrate.quad*.

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
