# Peer review of "Dualities and loops on squashed $S^3$"

_SciPost Physics, doi:SciPost Phys. 15, 156 (2023)_

## Round 2 · Referee Report · Anonymous (Referee 2) · 2023-3-15

Report

This paper studies 3d theories on the squashed sphere with extended supersymmetry (more than 4 supercharges). These theories/backgrounds were constructed previously and this paper focuses on finding BPS line operators and discussing the partition functions and expectation values of the operators.

Overall this is an original and rigorous paper that should be published. I have two main concerns that I'd like the author to address. One is a matter of terminology and the other is related to the numerical calculations.

The author refers to the theories as having ${\cal N}=4$ supersymmetry. This language attempts to separate the theory from the background (geometry, masses, fluxes). Such a split is unnatural in this context and indeed the relations shown for these theories are associated to changing the squashing and matching theories with different parameters. I would recommend to find a different language that recognizes that the field content is the same as ${\cal N}=4$ theories in flat space, but there is no $SO(4)$ R-symmetry (or eight preserved supercharges).

Also in the discussion of mirror pairs, the discussion should be more relations between partition functions and observables rather than theories. This is evident in the formulas, but not in the preceding discussions.

I am not very familiar with numerical calculations of such integrals and the paper does not explain it in much detail or give appropriate references. There are no error estimates and the differences of $10^{-5}$ or $10^{-3}$ are said to be good. A match is good if the difference is of order of the expected numerical error, but no error estimates of the numerics are provided and I think that fixing this will significantly strengthen this part of the paper

Requested changes

  1. A clearer language to refer to theories + backgrounds with enhanced supersymmetry.
  2. Error estimates of the numerical calculations.

---

## Round 2 · Referee Report · Anonymous (Referee 1) · 2023-3-15

Report

The author considers $\mathcal{N}=4$ supersymmetric quantum field theories on a squashed three-sphere. As in one of the previous works by the author and collaborators, a specific choice of the mass parameters is made such that 6 out of 8 supercharges are preserved. In such a setup, the author considers line operators that preserve 2/3 of the six unbroken supercharges. The author then argues that the expectation values of those operators can be expressed via the ones on a round (i.e. non-squashed) three-sphere. The author also considers numerical evaluations of the partition functions of two specific theories: ABJM and SYM, with various values of the parameters (such that four supercharges are preserved) and finds very strong evidence of their matching, which is in agreement with the conjectured duality between these two theories.

I believe that the obtained results and the developed techniques are of interest to other researchers working on related topics, such as supersymmetric localization and dualities. The paper is in general well-written and the presentation is clear. I recommend the manuscript for publication.

Requested changes

I have just very minor suggestions:

1) I think it could be helpful to clarify the notations $A$ and $\sigma$ appearing in (3.1). In particular, it is somewhat clear that A there is not the same as $A$ in section 2 (the gauge field in the supergravity multiplet), but it might be helpful to clarify it after (3.1) anyway.

2) In the sentence above (2.15) there is a typo: missing "h" in "the".

3) It could be helpful to add clarification of the meaning of the + operation in $\vec{\eta}/b+iq/b$ in the middle line of (4.4), as $q$, unlike $\vec{\eta}$, does not seem to be a vector.

---

## Round 3 · Referee Report · Anonymous (Referee 1) · 2023-8-25

Report

The author has made appropriate changes. I would like to recommend the manuscript for publication.

---

## Round 3 · Referee Report · Anonymous (Referee 2) · 2023-9-4

Report

The revision addresses the requested changes and I therefore recommend publication.

---

## Round 3 · Author Response

We thank the referees for their careful reading of the manuscript. We have included all of their suggestions in the new manuscript, a list of changes is given.

---

## Round 3 · List of Changes

1. Fixed figures 2(b) and 3(b) to show annotation of vertical axes
    1. Fixed several typos
    2. Made language on supersymmetry of backgrounds and gauge theories clearer throughout the whole manuscript: To the backgrounds we now refer only by the number of supercharges they preserve. To the gauge theories we refer by the amount of supersymmetry they have in flat space. Added a footnote in the introduction to clarify this point. This has affected in section 1 paragraphs 2, 3, 6, 9, in section 2 paragraph 1 as well as after equation (2.7) and before (2.11), in section 3 the paragraphs after equations (3.5) and (3.6) and in section 4.1 the ends of paragraphs 1, 2 and 3.
    3. Added a clarification on the fields in the defining equation (3.1) for the Wilson loop
    4. Added a clarification for the notation in the second line of equation (4.4)
    5. In section 4, made it more explicit that we are discussing the duality mapping at the level of partition functions and loop operators. The main changes are the first paragraph of the section as well as the captions to Figures 6 and 8
    6. Added a paragraph in subsection 4.2 on expected numerical errors
    7. Added comparison to the exact round sphere result on the full mass and FI parameter range with two additional figures
    8. In all of subsection 4.2, changed wordings to reflect the comparison to the expected errors

---

## Editorial Decision

published